# Electrochemically Produced Copolymers of Pyrrole and Its Derivatives: A Plentitude of Material Properties Using “Simple” Heterocyclic Co-Monomers

**DOI:** 10.3390/ma14020281

**Published:** 2021-01-07

**Authors:** Tomasz Jarosz, Przemyslaw Ledwon

**Affiliations:** Department of Physical Chemistry and Technology of Polymers, Silesian University of Technology, 9 Strzody Street, 44-100 Gliwice, Poland; tomasz.jarosz@polsl.pl

**Keywords:** pyrrole, polypyrrole, electrochemical polymerisation, copolymer, biosensing, sensor

## Abstract

Polypyrrole is a classical, well-known conjugated polymer that is produced from a simple heterocyclic system. Numerous pyrrole derivatives exhibit biological activity, and the repeat unit is a common building block present in the chemical structure of many polymeric materials, finding wide application, primarily in optoelectronics and sensing. In this work, we focus on the variety of copolymers and their material properties that can be produced electrochemically, even though all these systems are obtained from mixtures of the “simple” pyrrole monomer and its derivatives with different conjugated and non-conjugated species.

## 1. Introduction

Polypyrrole (PPy) and its derivatives are one of the most commonly investigated conducting polymers [1]. Moreover, both pyrrole and many of its derivatives are commercially available from various global vendors. PPy has attracted such interest due to its biocompatibility [2], high conductivity [3], and the possibility of producing it via electrochemical polymerisation in various solvents [4]. However, poor mechanical properties and lack of solubility in common organic solvents limit the applications of PPy [5]. These properties can be improved. One of the important strategies for achieving that is copolymerisation of pyrrole with other compounds. Copolymerisation is an important strategy of polymer synthesis which can lead to significant modification of desired properties compared to homopolymer. The properties of polypyrroles are dependent on both the polymerisation conditions and the chemical structure of the used monomers. Electrochemically produced PPy and its derivatives are substances that are interesting in terms of developing semiconductor materials with a broad array of applications, including medical [6,7], electronic [8], sensing devices [9,10], and energy storage devices [11]. Copolymers of pyrrole (Py), which were produced via electrochemical copolymerisation in a solution containing a mixture of different co-monomers, are one particular class of polypyrrole-based materials. The first work on this class of materials has been published in 1979 [12]. Investigations on such pyrrole copolymers are continued to this day, due to the constant emergence of new applications and usage aspects of those copolymers.

Electrochemical oxidative polymerisation was of interest to several research groups, which resulted in the various proposed reaction mechanisms [13]. This is due to the insolubility of the polypyrrole and problems with the separation of intermediates. Electrochemical polymerisation of pyrrole and electrochemical study of polypyrrole were studied in detail by Zhou and Heinze [14]. Their work shows that the properties of electrochemically obtained polymer films differ, depending on electrodeposition conditions, such as polymerisation potential [14] acidity of the pyrrole solution [15] and occurrence of the “water effect” [16]. The mechanism of electrochemical polymerisation involving multiple steps is depicted by Scheme 1 [17]. According to this, most widely accepted oxidative polymerisation mechanism of simple heterocyclic compounds, polymers with different chain lengths, cross-links, and hence different morphology and electrochemical properties can be formed [16].

Taking into account the complex nature of electrochemical polymerisation and significant influence of the employed process conditions, different copolymer structures can be expected in the case of different pyrrole co-monomers, as shown in Scheme 2. Based on the complex electropolymerisation mechanism, random [18] and block copolymers, graft copolymers, interpenetrating polymer blends [19], as well as complex systems of different products can be expected. This is reflected in the published works on pyrrole copolymers obtained not only by electrochemical, but also by chemical oxidative copolymerisation, in which copolymers with complex architectures were formed [20]. Moreover, the analysis of the molar ratio of different co-monomers in the starting solution and the composition of the copolymer is not identical, suggesting a significant impact of the polymerisation rate of co-monomers on the final structure [21,22].

The literature reviewed in this work has been divided into categories, based on whether the works focused on investigating the electrochemical copolymerisation of pyrrole-containing species or whether the works focused on the application of obtained copolymers.

## 2. Copolymerisation of Pyrrole

### 2.1. Copolymerisation of Pyrrole-Based Co-Monomers

The initial research on pyrrole copolymerisation involved the simplest pyrrole derivative it is N-methylpyrrole (**1**) (Figure 1) [18]. Electrochemically prepared semiconducting films, obtained via the electrochemical deposition of mixtures of Py and **1** present properties that are intermediate in regards to the properties of the homopolymers obtained from the two monomers. The conductivity of PPy is in the range 40–100 S∙cm^−1^, while the conductivity of poly(**1**) is considerably lower and approximately 10–3 S∙cm^−1^ [18]. The values of copolymer conductivity are interposed between values for the respective homopolymers and depend on the monomer ratio. Similar dependency was recorded for redox potentials, indicating that a random copolymer was formed. The presence of true copolymers instead of mixture of homopolymers has been also confirmed by IR spectroscopy study in other article [23].

The study of electrochemical copolymerisation of N-methylpyrrole (**1**) and 3-methylpyrrole (**2**) provides information about the influence of the polymer composition on the changes of the film properties in time [24]. IR spectra recorded for copolymer films obtained at different electrodeposition time show different content of **1** and **2**. This was reflected in the changes of the copolymer film properties. The comparison of homopolymers and related copolymers revealed that the conductivity of copolymer remains similar even after a year, whilst the conductivity of the homopolymers decreases faster. This is indicative of a significant polymerisation effect, beyond simply summing up the properties of the two copolymer parts.

Copolymers of different N-alkylpyrroles (**3**–**5**) and Py are another example of new electrochemically produced materials which slight change in the monomer structure leads to significant changes in properties [25]. Copolymers and homopolymers were synthetized via both electrochemical and chemical polymerisation protocols. Their electrical conductivity was studied in detail. The highest conductivity 0.54 S∙cm^−1^ was recorded for PPy. The increase in the alkyl chain length of the utilized N-alkylpyrrole co-monomer results in decreasing conductivity of the copolymer. A similar correlation was recorded for copolymers, with increasing the ratio of N-alkylpyrrole units to pyrrole units resulting in decreasing conductivity.

Electrochemical copolymerisation of Py and either **6** or **7** produces copolymer films with lower conductivity than pure PPy; however, the incorporation of 2-phenylethynyl groups originating from eight into the copolymer chain considerably enhances its thermal stability [26]. Moreover, the presence of triple bonds allows the copolymers to be subjected to photo-crosslinking.

Another studied group of copolymers features N-phenyl-substituted Py and their derivatives (**8**–**10**) (Figure 1) ([27,28]). The inclusion of even small amounts of co-monomers **8**–**10** strongly affects their properties, such as conductivity [21]. The addition of even 10% of either one of these co-monomers to the copolymerisation solution containing Py leads to the electrodeposition of copolymer films with conductivity lower by even six orders of magnitude. The differences in the oxidation potentials of the monomers were found to affect the difference in reactivity of monomers and hence copolymer properties. The elemental analysis reveals that the composition of the copolymer is not identical to the molar ratio of the starting solution [21]. This indicates different polymerisation rates of co-monomers. 

The substitution of the phenyl ring with thiol groups (**11**,**12**) is an interesting modification, aimed at granting the compounds self-assembly properties, on gold surfaces in particular [29]. Once monolayers of these compounds on gold electrodes were prepared, they were oxidized, after which pyrrole was added into the system and copolymerisation was carried out. Interestingly, the choice of **12** and **13** as the co-monomers served to greatly increase the rate of polymerisation (up to 70-fold increase) and simultaneously lower the capacitance of the polymer/solution interfaces and charge transport resistance values, in comparison with pure PPy on gold.

The electrochemical copolymerisation of Py and **13** was found to lead to a self-doped electroactive copolymer, in which counter-ions are covalently attached to the polymer chain [30]. This type of copolymers can be used as water soluble materials for cation specific membranes [31].

The copolymerisation of pyrrole with **14** is reported to yield an uncommon copolymer, containing pyrrole and vinylpyrrole segments in its main chain [32]. This copolymer was found to have a helical cone-like morphology, possibly useful in future sensing applications, and showed both higher conductivity and better thermal stability than PPy. These improved properties results from the enhanced conjugation in the copolymer main chain.

The electrochemical galvanostatic copolymerisation of Py with N-hydroxyalkylpyrroles **15**, **16** leads to copolymers with high charge/discharge capacity [33]. Despite the higher oxidation potential ascribed to the steric hindrance of these monomers in respect to Py, the resultant copolymers present redox features similar to those of PPy. Moreover, the proper quantity of these groups enhances the ionic mobility of the PPy. SEM analysis revealed that during the electropolymerisation the homopolymers grew two-dimensionally while the copolymers grew three-dimensionally.

Co-monomers **8**,**17**–**19**, as well as the pentafluorophenol ester of **18** were used to electrochemically produce copolymers with Py [22]. A non-linear correlation between solution and copolymer composition was observed, with the ratio between Py and co-monomer units being dependent on the co-monomer structure and its reactivity in comparison with Py. The increase of monomer size resulted in reduction of functionalized units in the copolymer. Pyrrole and the carboxyl-functionalized N-alkyl pyrrole **18** were used for copolymerisation at the air/liquid interface under the control of an electrical field [34]. Copolymer film grew along the direction of the electrical field. The morphology studies show completely different surface of the polymer from air and liquid directions.

Another example of copolymerisation is the electrochemical deposition of **18** and its butyl ester **20** [35]. In this study the effect of nature of the hydrophilic **18** and hydrophobic **20** on the mechanism of electrochemical deposition was studied. Measurement reveals that **18** undergoes progressive nucleation in acetonitrile and then instantaneous growth of the **20** nucleates occurs. Changing the conditions of electrodeposition and co-monomer ratio results in polymer films differing in terms of their structural, chemical, and electrical properties, but typically the copolymers are enriched in units originating from **20** in comparison with the polymerisation solution composition 

The electrochemical copolymerisation of Py with a star-shaped Py-functionalized monomer **21** (Figure 2) results in copolymers with probable cross-linked structure [36]. Copolymers have narrower band gaps and improved conductivity, when compared to pure PPy. Those systems have advantageous properties which can be beneficial in some applications such as electrochromic devices.

The electrochemical copolymerisation of PPy and PPy-functionalised dendrimer were used to obtain a dendritic conducting star copolymer poly(propylene imine)-co-PPy, exhibiting an altogether different morphology than PPy (Figure 3). Electrochemical impedance data confirmed that the star copolymer is a semiconducting material [37]. 

Another example is the electrochemical oxidation of Py in the presence of **22** [38]. Such a co-monomer structure results in cross-linked copolymer architecture, leading to polyether/polypyrrole cages. The temperature dependence of dark electrical conductivity of obtained copolymer indicates that charge carrier transport is dominated by the thermal excitation. Similar cross-linked structure was obtained by the electrochemical deposition of **23** and Py copolymers [39]. Those copolymers were proposed as conducting materials for batteries, sensors and metal complexing reagents.

Electrochemical oxidation was used in copolymerisation of Py with styrene and functionalized polystyrene. The first example is the direct electrochemical oxidation of monomers: PPy and styrene in solution. Surprisingly, the electrochemical copolymerisation led to the block copolymer not a composite or a blend [40]. The electrochemical oxidation of Py with a Py-substituted polystyrene (**24**) is another example [41]. PPy grafted on the polystyrene chains are formed as a result of a constant potential electrolysis. The detailed SEM and conductivity measurement were made to characterize copolymers. Another example includes the electrochemical oxidation of Py and polystyrene with terminal Py groups (**25**) [42]. Such copolymerisation leads to a copolymer with good electrochemical stability. 

Electrochemical copolymerisation was used to investigate the role of incorporation of ferrocene derivatives as electroactive substituents (**26**). Initial studies of ferrocene incorporation into PPy films show that this can be controlled by the solution composition [43]. Further studies show that polymer films maintain the redox properties of incorporated electroactive groups. This property can be used in electrocatalysis, sensing and photoelectrochemistry. Obtained results show that β-substitution has less of an effect on PPy electroactivity than N-substitution (**26**–**28**) [44].

To summarize this chapter, although a great variety of co-monomers have been employed (Table 1), in very few cases does research dare depart from the use of Py as a basis for copolymerisation, likely owing to its rapid electrochemical polymerisation, due to the lack of any steric hindrance, unlike for its derivatives.

### 2.2. Copolymerisation of Pyrrole with Non-Pyrrole Comonomers

#### 2.2.1. Copolymerisation of Pyrrole with Thiophene Derivatives

Other conjugated species are an important group of co-monomers for electrochemical copolymerisation with pyrroles. Among them are mainly typical compounds used in conjugated polymer chemistry, such as furan, thiophene, aniline, and carbazole.

Py and thiophene (**T1**) (Figure 4) derivatives are one of the most commonly investigated combinations of co-monomers for electrochemical polymerisation. In their copolymerisation, a large excess of thiophene must typically be applied, due to significantly higher oxidation potential in comparison to that of pyrrole [45]. The pyrrole and thiophene ratio in the polymer film can be easily changed by changing the employed oxidation potential. This method was used in numerous papers, e.g., to prepare copolymers in the template-synthesis with Th-Py films deposited in the pores of microporous anodic aluminium oxide [46,47].

The electropolymerisation conditions were studied in the case of Py/bithiophene (**T2**) monomers [48]. **T2** has significantly lower oxidation potential than thiophene from this reason is better co-monomer. However, the difference of oxidation potential is approximately 0.3 V. Electrochemical impedance spectroscopy indicates the formation of a thin layer of PPy at the beginning of electrodeposition on electrode surface and then thicker true copolymer film formation. This was due to the lower oxidation potential of Py and significantly higher reactivity of radical cations of Py than that of **T2**.

IR spectroscopy and XPS were used to estimate the copolymer qualitative and quantitative data of the products of the copolymerisation of **29** (Figure 2) with **T2**–**T4** (Figure 4), **30** and **31**, as well as the copolymerisation of **32** with **T2**–**T5** and **31**. Results show that Py units are generally incorporated into copolymers in larger proportions than the thiophene units. In this study there is a linear correlation between co-monomers concentration in solution and copolymer composition [49].

Electropolymerisation of Py and **T6** results in a copolymer with higher capacitance than homopolymers up to 291 F·g^−1^. After 1000 charge/discharge cycles only a 9% decrease of capacitance was observed [50]. The copolymer electrode also presents an improved electrochemical stability.

#### 2.2.2. Copolymerisation of Pyrrole with Other Heterocyclic Compounds

Carbazole derivatives (**Cz1**–**Cz3**) are important group of conjugated compounds which were copolymerized with Py. Such copolymerisation provides information about the relation between structure of monomers and copolymers properties [51]. When copolymerisation takes place via the **C2** and **C5** positions of the pyrrole ring and the **C3** and **C6** positions of carbazole, copolymers with high conductivities can be obtained. Otherwise, less conductive forms are obtained. Other derivatives of those monomers (**Cz1**) were also electrochemically copolymerized with the formation of electroactive layers, for which detailed analysis by electrochemical impedance spectroscopy was made [52].

An interesting case is the electrochemical copolymerisation of pyrrole and indole (**In**), supposedly yielding poly(Py-co-indole) derivatives [53,54,55,56]. Pyrrole and indole can be electrochemically copolymerized owing to the similar onset oxidation potentials, whose difference does not exceed 0.1 V. The composition of electrodeposited copolymer depends on the co-monomer concentration ratios in the polymerized solution. Different mechanisms for the formation of polyindole/PPy copolymers were proposed. For the elucidation of these mechanisms, electrochemical, spectral and spectroelectrochemical techniques were used; however, it is still not fully understood, especially the incorporation of indole units. Obtained copolymers present intermediate spectroscopic properties between PPy and polyindole. The resistivity measurements show a lower conductivity of these copolymers than the corresponding homopolymers.

Electrochemical copolymerisation of pyrrole with furan in boron trifluoride ethyl etherate leads to copolymers with a large ratio of furan to pyrrole units, due to large difference of onset potential between monomers [57].

Copolymers of Py with aniline (**An1**) and their derivatives were reported. Detailed studies of electrochemical copolymerisation show that the choice of oxidation conditions and medium affects the resultant product properties [58]. Electrical conductivity measurement and UV-VIS spectroscopy were used to confirm the formation of a pyrrole/**An2** copolymer [59]. Direct electrochemomechanical deformation studies and cyclic voltammetry at low pH reveal that such copolymers show enhanced electrochemomechanical deformation and better electroactivity in those conditions in respect to their parent homopolymers. The copolymerisation of **An3**, containing a carboxyl substituent leads to self-doped copolymers [60]. Moreover, the presence of carboxyl groups makes it an attractive material for sensor applications. The copolymer exhibits crystallinity, with a tubular or fibrillar elongated structure, intermediate between the two parent homopolymers [61].

The electrochemical synthesis of a biphenyl-Py system was performed at the minimum potential, at which copolymerisation could take place [62,63]. The large difference between the oxidation potentials of pyrrole and biphenyl, which are 0.80 V and 1.80 V (vs. SCE), respectively, did not prevent copolymerisation, even when a potential of 0.8 V was applied for the process. This was confirmed by FTIR spectroscopy, X-ray diffraction analysis, TGA analysis and by the difference of solubility of homopolymers and copolymers in various solvents.

The electrochemical deposition of Py and phenol in the presence dodecyl benzene sulfonic acid (DBSA) with oxalic acid solution were made on steel substrates [64]. Different solution composition was studied. The morphology of the copolymers depends on the initial solution mixture particularly the amount of doped sulphur.

Imidazole (**Iz**) is an interesting choice of co-monomer for Py, as it is more frequently associated with n-dopable units, such as triazoles, oxadiazoles, and tiadiazoles. Although not as different from pyrrole, it was found to inhibit electrochemical polymerisation and result in a copolymer, whose morphology was significantly different than would be expected for PPy [65]. This morphology, featuring a well-developed surface area leads to it exhibiting supercapacitive properties, with a specific capacitance of 201 F·g^−1^ at 10 mV·s^−1^.

#### 2.2.3. Copolymerisation of Pyrrole with Other Compounds

The studies of electrochemical oxidation of propylene oxide show that this monomer cannot be polymerized in the conditions usually used for deposition of conjugated polymers; however, the electropolymerisation of Py in the presence of propylene oxide results in the formation of copolymer with PPy main chain and grafted poly(propylene oxide) side chains. This indicates the polymerisation of propylene oxide initiated by radical cations formed at PPy chains [66]. Similar direct electrochemical copolymerisation of PPy and a non-conjugated monomer was also reported for ε-caprolactone [67] and tetrahydrofuran [68].

Electroctropolymerisation of Py and an ionic liquid monomer was carried out in order to obtain PPy with improved electrochemical properties [19]. The detailed analysis of products indicates that a blend of homopolymers was formed rather than a copolymer. The polymer film has mixed electronic and ionic conductive properties. 

To summarize Section 2.2, different pairs of co-monomers used in literature have been compiled in Table 2. Comparing the electropolymerisation of pyrroles and non-pyrrole co-monomers with pyrrole-derivative co-monomers, a greater variety of polymer structures is noticeable. These structures include not only the random and block copolymers, but also graft copolymers and interpenetrating blends.

## 3. Application of Pyrrole Copolymers

### 3.1. Copolymers for Sensing Applications

PPy are one of the most important conjugated polymers for use in electrochemical sensors [69]. In this subsection, the examples of electrochemical copolymerisation for applications in sensors are collated. Due their significance, pyrrole-based materials for bio-sensing applications are reviewed in the next subsection.

Electrodeposition of PPy and Ni(II) salen complexes lead to formation of reproducible layers [70]. Obtained deposits display the characteristics of both components of the polymer film. The polymer composition was confirmed by spectroelectrochemical UV–Vis-ESR measurement and MALDI-TOF mass spectrometry. Such material is an interesting candidate for use in sensing applications, such as gas or heavy metal sensors.

Electropolymerisation of PPy-substituted vitamin B12 with PPy results in corrin-doped PPy film [71]. Different electropolymerisation parameters were investigated such as the concentration of the monomer, potential range, scan rate, and number of scans. These studies lead to the new thiocyanate-selective electrode with the long lifetime of at least 6 months with no significant deterioration in the slopes during this period.

Electropolymerisation of (D,L)-N-(1,2-dicarboxyethyl)-1H-pyrrole with pyrrole prepared from D,L-aspartic acid leads to polymers with unusual properties [72]. This polymer forms complexes with Cu^+^ and Cd^2+^ ions under an applied electric field. After changing the potential to 0 V, the release of the metal ions was observed. These properties indicate potential applications of the copolymer in the analysis of heavy metals and catalysis.

The electrochemically generated copolymer with an **18** content of 5% was used for fabrication of ethanol sensors [73]. In this purpose alcohol dehydrogenase was attached to the PPy copolymer deposited at electrode surface through amide linkages. So prepared microelectrodes were used for the amperometric sensing of ethanol.

An uncommon choice of co-monomer for sensing application is that of substituted styrenes [74]. The pyrrole-styrene copolymer sensors have been produced electrochemically in nitromethane and exposed to vapours of water and various organic solvents (toluene, dichloromethane, methanol, and acetonitrile). Interestingly, by utilizing sensors bearing different copolymers together, principal component analysis was enabled, allowing distinguishing between five analytes, and only failing to distinguish between dichloromethane and toluene.

### 3.2. Copolymers for Bio-Sensing Applications

One of the most intensively studied group of pyrrole copolymers are those intended for biomedical applications and primarily for bio-sensors. This follows from the biocompatibility of these materials, as well as their potential for sensing a wide variety of analytes, depending on the employed copolymer and its deposition and post-treatment. The biocompatibility of PPy with nerve tissue was confirmed according to international standards (ISO 10993 and ASTM F1748–82) [2]. However, further studies on PPy biocompatibility show that only low concentrations of PPy nanoparticles are biocompatible, while high concentrations can lead to cytotoxic effects [75].

The electrical interactions of conjugated polymers with biological systems can be used both for cell stimulation and recording bioelectrical signals (Figure 5). Electrodeposited polymers of pyrrole and dopamine were investigated for these purposes, as coating materials for electrodes [76]. In comparison with PPy-coated electrodes, polydopamine-Ppy coated electrodes exhibited both improved electrical and cell-supporting properties, the latter being evidenced by a significant improvement in the growth and differentiation of C2C12 myoblasts and PC12 neuronal cells on copolymer-coated electrodes.

Conjugated copolymers are great candidates for application in enzyme biosensors own to the possibility of combining of redox properties of conjugated polymers and capability of electrochemical copolymerisation of functionalized monomers, e.g., terpolymerisation of pyrrole-functionalized osmium complex with Py or 1 and Py-modified glucose oxidase lead to amperometric biosensors for glucose [77]. This application is based on the direct electrical communication of the enzyme with the electrode via the conducting polymer. Further studies of such pyrrole-bearing osmium complexes were carried out to solve some issues of the electrochemical generation of copolymers such as different oxidation potentials and formation of reproducible films [78]. As a solution potentiostatic and galvanostatic pulse-deposition methods were proposed. To improve the sensitivity, selectivity and stability of glucose sensors the modification of osmium-complex PPy were tested [79]. A longer methylene moiety between osmium complex and PPy units improves the glucose sensor performance own to the higher stability of obtained copolymers.

Another example is poly(PPy-biotin) for the immobilization of enzymes. PPy films were obtained by electrochemical oxidation of PPy linked to polypyridinyl complex of ruthenium(II) with the PPy linked to biotin derivative. Obtained copolymers were used as receptor layers for glucose and phenylphospate biosensors [80]. The copolymerisation of PPy ruthenium complex with 3-methylthiophene also lead to polymer film with active redox Ru(II/III) system with possible sensor application [81].

The “macromonomer” technique via electrochemical deposition of Py and AzbPy-g-PEG was used to obtain copolymer presented at Figure 6 [82]. Owing to the presence of PEG side chains the enhanced biocompatibility of the copolymer in comparison to PPy was recorded. These copolymers were tested in implantable electrodes for serotonin detection revealing promising properties.

Electrochemically generated copolymers of PPy with 1-(2-carboxyethyl)pyrrole were used to obtain Glucose oxidase immobilized conducting film [83]. The various content of Py-COOH units lead to change in amperometric response and film conductivity. The Py-COOH units played an important role in Glucose oxidase immobilized on conjugated polymer surface, simultaneously an increase content of these units cause a decrease in the conductivity and hence the sensing ability. These studies show that appropriate ratio between Py and Py-COOH play a key role for these applications.

Copolymers of vinyl alcohol with thiophene side groups and PPy electrochemically polymerized were used for the immobilization of invertase and glucose oxidase [84]. The influence of electrolysis conditions on enzyme activity was studied.

The direct copolymerisation of PPy and covalently linked oligonucleotides at C5′ of PPy is a method for the electrochemical addressing of DNA [85]. This technique employs electrochemical copolymerisation of these monomers at microelectrodes. This leads to the elaboration of biosensors. Similar studies explored conducting polymers as a route for localizing DNA onto conducting surfaces [86]. To exemplify, PPy-COOH and its succinimidyl ester were electro-copolymerized at the electrode surface. Further studies include attachment of amine-terminated DNA to copolymer and their investigation [87]. Other works on DNA sensing include the use of different PPy derivatives with carboxylic groups such as 3-pyrollylacrylic acid.

Generally in these studies of DNA sensing by electrochemical approaches PPy copolymers are used as linkers between a substrate and oligonucleotide probes [88].

Similar reports include the protein sensing. For example electrochemical copolymerisation of PPy-modified protein and PPy is an efficient grafting process which immobilizes molecules at defined positions on a gold surface [89]. This technique can be used for sensing of various proteins by detecting different protein-protein interactions. Poly(PPy-co-pyrrolepropylic acid) film electrochemically deposited on indium-tin-oxide and bio-functionalized with myoglobin protein antibody. The bioelectrode studies in a phosphate buffer solution of detection of myoglobin protein antigen exhibited a linear impedimetric response to the protein concentration [90]. Copolymers of these monomers were also tested with Goat IgG as a model protein surface plasmon resonance [91]. An oligosaccharide array to measure the carbohydrate-binding protein interactions were made by electrocopolymerisation of the PPy with oligosaccharides linked covalently to PPy above a gold surface. This enables the covalent immobilization of multiple probes [92].

Electrochemical copolymerisation of Py with Py propylic acid was used to obtain conjugated polymers with absorption of proteins [93]. The FTIR spectra confirm the copolymer formation. Detailed experimental indicated that studied copolymer have a more hydrophilic and negatively charged than PPy surface and hence the content of Py propylic acid units influence the absorption behaviour of proteins on the prepared film.

An amperometric biosensor was also used for urea determination [94,95]. For this purpose, a new copolymer poly(N-3-aminopropylpyrroleco-PPy) was electrochemically obtained. In these applications, the enzyme urease was covalently immobilized to the electrode surface coated by this copolymer. An amperometric or potentiometric response was measured as a function of urea concentration.

Tyrosinase, enzyme called as poly phenol oxidase can be covalently immobilized on a copolymer of N-3-aminopropyl and PPy [96]. This type of material with covalent linkage of enzyme to polymer chain and porous morphology lead to the high enzyme loading and a good stability of the enzyme electrode. Another example is immobilization of tyrosinase in PPy film by copolymerisation of a polydimethylsiloxane with a Py end group with Py in the presence of tyrosinase [97]. In these copolymerisation processes, the enzyme is entrapped in conducting copolymer. After condition optimization, electrodes with entrapped enzyme were used for the determination of phenolic contents in green and black.

An interesting ethanol bio-sensing solution was developed by immobilizing alcohol oxidase enzyme on the surface of an electrode coated with a copolymer of pyrrole-3-carboxylic acid and a dithienylbenzoselenadiazole [98]. The use of this selenoorganic co-monomer was dictated by its biocompatibility and ability to improve charge transport, which is highly relevant for sensors utilizing immobilized enzymes. The resultant ethanol sensors showed linearity in a wide range of concentrations and a very low detection threshold of 0.052 mM ethanol.

Immobilized enzymes (β-galactosidase and galactose oxidase) were also employed for the detection of lactose in milk, with a copolymer of Py and 3,4-ethylenedioxythiophene (EDOT) being used as the immobilizing matrix [99]. The produced bio-sensors were found to respond rapidly (response time of 8–10 s) and featured an excellent detection threshold of 0.014 mM lactose.

Another bio-sensor of this sort, utilizing an immobilized fusion protein consisting of the lectin concanavalin A and streptavidin, was employed for the detection of antibodies [100]. Several pyrrole-based copolymers were investigated as the immobilizing matrices for surface plasmon resonance measurements, with copolymers of pyrrole with N-octadecylpyrrole, as well as pyrrole with pyrrole-N-hexanoic acid showing the most favourable properties.

A fairly recent approach to developing bio-sensors is via molecular imprinting of the analyte in a conducting polymer matrix [101]. Such a paracetamol bio-sensor has been reported using an electrochemically polymerized Py-carbazole copolymer matrix. Although an extremely complex electrode–glassy carbon modified with AuPd alloy nanoparticles, graphene-carbon nanotube, and an ionic liquid—the resultant sensor was found to have decent performance and a low detection threshold of 50 nM paracetamol.

D-dimer is a blood-borne biomarker for the occurrence of deep vein thrombosis and pulmonary embolism [102]. This substance can be detected at a ng/mL concentration level using a microfluidic bio-sensor based on a copolymer of Py and N-hydroxyphtalimidepyrrole. 

### 3.3. Copolymers for Electrochromic Applications

Electrochromism is the phenomenon, in which a material changes its colour when voltage is applied to it. This colour (chromic) change typically is associated with the occurrence of reduction-oxidation reactions and can be observed for inorganic species, such as Ag nanoparticles [103] as well as for organic semiconductors, such as copolymers of pyrrole [104].

The electrochromic properties of a particular electroactive material are referred to by the colours that can be achieved and which typically correspond to stable redox states of the electroactive material, e.g., transparent-to-black electrochromism, which in itself is a material feature attracting significant research interest [105]. Materials exhibiting more than two coloured states are typically referred to as multielectrochromic and can exhibit e.g., a yellow-violet-black colour transition. The use of pyrrole derivatives and their copolymers for electrochromic devices has been the subject of an extensive and still fairly current review [106].

Multielectrochromism was reported for copolymers of different PPy derivatives with 3,4-ethylenedioxythiophene (EDOT). EDOT significantly improves the electroactive stability of PPy derivatives and for different systems leads to better electrochromic properties in different oxidation states [107]. This type of copolymers, depending on their structure, can exhibit multielectochromic behaviour involving up to six colours (Figure 7) [108,109]. The combination of bithiazole units with N-substituted PPy in one monomer and electrochemical copolymerisation with EDOT lead to the electrochromic material with yellow to grey transition during oxidation. The copolymers revealed very low switching time 0.6 s, high optical contrast up to 54% and good stability under atmospheric conditions [110].

The effect of monomer ratio of Py and 3,4-ethylenedioxythiophene (EDOT) on homogeneity of resulting copolymer have also reported. The most homogeneous mixture of the components were observed for high EDOT mole fraction exceeded 0.9 [111]. The copolymerisation of PPy and 3,4-ethylenedioxythio in an ionic liquid 1-butyl-3-methylimidazolium tetrafluoroborate and their electrochromic behaviour were studied. The in situ spectroelectrochemical measurement of PPy-co-PEDOT (2:1) reveals transitions from neutral to polaron and bipolaron states. In dependence of monomer ratio different colour were observed [112]. The electrochemical copolymerisation of these monomers was also studied in aqueous micellar solution. Different electrochromic properties of polymer films electrodeposited on ITO electrodes were observed with various monomer ratios for example with different solution composition results in different film colours under various potentials. Moreover voltammetry measurements indicate the copolymer film better stability than homopolymer with preserved electroactivity even after 1000 cycles [113].

A variety of block copolymers has been reported—the copolymers are based on Py and other aromatic monomers linked with insulating polymer segments, such as poly(methyl methacrylate) [114] or poly(ethylene oxide) [115] linked to a thiophene or polytetrahydrofuran [116] and poly(2-methyl-2-oxazoline) [117] linked to a Py end group. These molecules were electropolymerised with Py via constant potential electrolysis. These polymers were studies by spectral and electrochemical techniques revealing electrochromic behaviour. This research was aimed at optimizing the parameters of these copolymers such as sufficient electrochemical parameters and solubility.

### 3.4. Copolymers for Anticorrosive Applications

Homopolymers and copolymers of pyrrole and **1** were investigated as mild steel anticorrosive protection coating. The best results were obtained for copolymer films obtained with a 3:1 concentration ratio of Py:**1**. These copolymers exhibit better barrier properties and higher durability than either of their parent homopolymers, with the effect being ascribed to the hydrophobicity-promoting N-methyl substituent. However, the increased content of one, simultaneously decreases the adsorption strength of the copolymer to the substrate and at higher contents worsens the protective features [118]. These copolymers were also tested as copper protective materials. In this case, the best results were obtained from a 4:1 co-monomer concentration ratio [119].

The electrochemical copolymerisation of pyrrole and **An4** (Figure 4) was employed to produce good quality coatings on steel, with pyrrole being employed as a co-monomer because the parent **An4** homopolymer is not stable on either platinum or mild steel electrodes. It was found that electrochemically deposited copolymers are an effective coating against the corrosion of mild steel with high stability and low permeability in solutions of aggressive media [120]. The morphology and structure characterization by SEM, FTIR, and EDS reveals some similarity in the structure of the parent homopolymers and the copolymer but differs in terms of the size of the deposited particles [121]. TAz and pyrrole copolymers were also studied as promising conjugated polymer for applications in corrosion protection for mild steel in acid medium. The detailed electrochemistry and kinetic studies indicate block structure of electrogenerated copolymer with higher pyrrole content than that employed in the co-monomer mixture [122].

### 3.5. Other Applications

The possible application of PPy copolymers includes many others possible applications. In this subsection examples of other applications are listed. 

Electrochemically generated PPy copolymers with oligo(ethyleneoxy) chains at the 3-position, such as **34** and **35** (Figure 2) were tested as electroactive materials for battery electrodes. Copolymerisation lead to good quality polymer films with high conductivity up to 100 S·cm^−1^ and high stability. The good charge and discharge stability and large current density indicate a high potential of electrochemical copolymerisation of these group of PPy for use as active materials in battery electrodes [123].

The applied potential, the composition of the co-monomer solution and the film thickness of the Py and **36** were investigated in order to obtain copolymer with specifically functionalized nanotubes with potential application in drug delivery [124]. The immobilization with poly(ethylene glycol) chains, a protein-repellent polymer enhances the antifouling properties of these polymers.

## 4. Conclusions

Although pyrrole is a relatively simple heterocycle, by matching it with various systems, even as similar as C- or N-substituted pyrroles, copolymers exhibiting an extremely wide array of properties and tailored to a variety of applications can be produced. Among the many cited works, pyrrole has been coupled with both conjugated and non-conjugated co-monomers and attracted interest in terms of both basic and applied research.

Among the application-oriented works, bio-sensing and medical applications are the most common, due to the reported biocompatibility of PPy. This is the case, even despite some questions as to the limitations of this biocompatibility being raised. Even so, pyrrole-bearing copolymers can also be used for “classical” sensing, for the detection and distinguishing between both organic and inorganic species.

In the near future, the abovementioned trend in not expected to change significantly. The development of pyrrole-based biomaterials may, however, be stunted if a more in-depth study of the biocompatibility of PPy and its derivatives produces unfavourable results.

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
