# Peer review of "Electrochemically Produced Copolymers of Pyrrole and Its Derivatives: A Plentitude of Material Properties Using “Simple” Heterocyclic Co-Monomers"

_materials, 2021, doi:10.3390/ma14020281_

Round 1

Reviewer 1 Report

My comments and suggestions for authors can be found in the attached file.

Author Response

  1. Copolymerisation of pyrrole, 2.1 Copolymerisation of: On lines 43-44 it is stated that the SEM micrographs reveal fundamental differences between the morphology of the homoand copolymers. The used dopant ion during the polymerization of these films should be stated here. The morphology is highly dependent on the used counter ion, dopant ion. Was it the same for both the homo- and copolymers?

Respond:  PPy and poly(N-methylpyrrole) were prepared under the same conditions in a lithium perchlorate-containing acetonitrile electrolyte solution. In the revised manuscript this paragraph was rewritten without reference to SEM measurement (lines 80-82).

  1. Copolymerisation of pyrrole, 2.1 Copolymerisation of pyrrole-based co-monomers: The content of the 4th paragraph starting with “Homopolymers and copolymers pyrrole and 1…” should be moved to the end of the 1st paragraph.

Respond: I agree that this paragraph should be placed elsewhere. We reorganized the manuscript structure. This paragraph has been moved to the new paragraph: 3.4 Copolymers for anticorrosive applications. (lines 499-507)

  1. Copolymerisation of pyrrole, 2.1 Copolymerisation of pyrrole-based co-monomers: The reasons behind the improved conductivity in the copolymers of Py with a star-shaped Pyfunctionalised monomer (6) should shortly be discussed on paragraph 5.

Respond: The properties are probably due to the cross-linked structure. The paragraph has been improved: “The electrochemical copolymerisation of Py with a star-shaped Py-functionalised monomer 21 (Fig. 2) results in copolymers with probable cross-linked structure [36]. Copolymers have narrower band gaps and improved conductivity, when compared to pure PPy. Those systems have an advantage properties which can be beneficial in some applications such as electrochromic devices.” (lines 153-156)

  1. Copolymerisation of pyrrole, 2.1 Copolymerisation of pyrrole-based co-monomers: On line 112 the numbers for the Py co-monomers studied should not be in parenthesis.

Respond: It was icorrected: “IR spectroscopy and XPS were used to estimate the copolymer qualitative and quantitative data of the products of the copolymerisation of 29 (Fig. 2) with T2-T4 (Fig. 4), 30 and 31, as well as the copolymerisation of 32 with T2-T5 and 31.” (lines 219-221)

  1. Copolymerisation of pyrrole, 2.1 Copolymerisation of pyrrole-based co-monomers: On line 115 it should be specified that the meant comonomer (19) is the carboxyl functionalized N-alkyl Py. The text in this paragraph should also be moved to the 12th paragraph which starts by “X-ray photoelectron spectroscopy (XPS)…”.

Respond: It was speciefied: “Pyrrole and the carboxyl-functionalised N-alkyl pyrrole 18 were used for copolymerisation at the air/liquid interface under the control of an electrical field [34].”

The indicated paragraphs have been combined. (lines 137-145)

  1. Weather the used co-monomers of Py are commercially available could be a nice addition into the manuscript.

Respond: We added the sentence to the Introduction Section : “Moreover, both pyrrole and many of its derivatives are commercially available from various global vendors.”  (lines 21-22)

  1. Copolymerisation of pyrrole, 2.2 Copolymerisation of pyrrole with other precursors of conjugated polymers: The paragraph under “Copolymerisation of pyrrole-based comonomers” starting by “The FT-IR and XPS techniques were used to estimate…” should be moved to be under the title 2.2 because it discusses copolymerization of Py with bi- or terthiophenes.

Respond: This paragraph has been moved to subsections:

2.2 Copolymerisation of pyrrole with non-pyrrole comonomers

2.2.1 Copolymerisation of pyrrole with thiophene derivatives

(lines 219-223)

  1. Copolymerisation of pyrrole, 2.2 Copolymerisation of pyrrole with other precursors of conjugated polymers: On the 3rd paragraph, it should be discussed what were the reasons behind the formation of a thin layer of PPy at the beginning of the polymerization and then a thicker copolymer film on top of that.

Respond: The explanation was added: “This was due to the lower oxidation potential of Py and significantly higher reactivity of radical cations of Py than that of T2.” (lines 217-219)

  1. Copolymerisation of pyrrole, 2.2 Copolymerisation of pyrrole with other precursors of conjugated polymers: Correct “homogeneous polymer” to homopolymers on line 172. On line 176, correct “…their derivatives...” to its derivatives.

Respond: it was corrected (line 245)

What is meant by “electronic absorption spectroscopy” on lines 178-179?

Respond: It has been changed to UV-VIS spectroscopy (line 251)

Define what is meant by “electrochemomechanical deformation” on lines 180 and 181?

Respond: Electrochemomechanical deformation measurement is described in ref.: S. S. Pandey, W. Takashima, K. Kaneto, Electrochemomechanical deformation in conducting 549 copolymers containing pyrrole and anisidine moieties, Reactive & Functional Polymers, 2004, 58, 103–550 110

Correct the spelling of “carboxyllic” on lines 183 and 184.

Respond: It was corrected to carboxylic (lines 155-156)

Correct ”…morphology characterisation by SEM, FTIR and EDS” on line 192. FTIR cannot be used to study the morphology.

Respond: We added structure characterisation: ”The morphology and structure characterisation by SEM, FTIR and EDS….” (line 512)

On line 200 move “respectively” to be after the given potential values.

Respond: It was corrected (line 261)

Rewrite the following sentence on lines 204-206 “The studies of electrochemically deposited on steel substrates copolymers of Py and phenol 205 with different solution composition in the presence dodecyl benzene sulphonic acid (DBSA) with oxalic acid solution were made.”

Respond: It was rewritten: ” The electrochemical deposition of Py and phenol in the presence dodecyl benzene sulphonic acid (DBSA) with oxalic acid solution were made on steel substrates [64]. Different solution composition was studied. The morphology of the copolymers depends on the initial solution mixture particularly the amount of doped sulphur.” (lines 265-288)

  1. The chemical structures of all presented co-monomers other that including Py should also be shown in a figure.

Respond: The monomer structures without pyrrole are shown in Figure 4

  1. Copolymerisation of pyrrole, 2.3 Copolymerisation of pyrrole with non-conjugated species: Rewrite the following “The smooth charge and discharge and good time of supplying the current…” on line 218.

Respond: The sentence was rewrited: “The good charge and discharge stability and large current density indicate a high potential of electrochemical copolymerisation of these group of PPy for use as active materials in battery electrodes” (lines 525-527)

What method is meant in the following sentence “In literature this method was also used to obtain…” on line 226.

Respond: The sentence was rewritten: “The electrochemical copolymerisation of PPy and PPy-functionalised dendrimer were used to obtain a dendritic conducting star copolymer poly(propylene imine)-co-PPy, exhibiting an altogether different morphology than PPy (Fig. 3).” (lines 165-167)

 Correct “Surpassingly” on line 243.

Respond: It was corrected (line 178)

  1. The reasons behind the chosen co-monomers to be polymerized with Py in the studies presented in this review would be a good addition.

Respond: The explanation is in the Introduction Section: “However, poor  mechanical properties and lack of solubility in common organic solvents limit the applications of PPy [5]. These properties can be improved. One of the important strategies for achieving that is copolymerisation of pyrrole with other compounds. Copolymerisation is an important strategy of polymer synthesis which can lead to significant modification of desired properties compared to homopolymer. The properties of polypyrroles are dependent on both the polymerisation conditions and the chemical structure of the used monomers.” (lines 24-30)

  1. I could not find any mention to Figure 5 in the text. Check this point.

Respond: All figures were highlighted in the manuscript text.

  1. Application of pyrrole copolymers, 3.1 Copolymers for sensing applications: In my mind the polymers resultant from polymerization of Py in the presence of electroactive substituents such as Ni(II) salen complexes cannot be called copolymers. They are rather composites of polypyrrole. Define what slopes are meant in “…deterioration in the slopes…”. Define what is meant by ”–ve” on line 278.

Respond: This part was modified considering the above doubts: “Electrodeposition of PPy and Ni(II) salen complexes lead to formation of reproducible layers [71]. Obtained deposits display the characteristics of both components of the polymer film. The polymer composition was confirmed by spectroelectrochemical UV–vis-ESR measurement and MALDI-TOF mass spectrometry.” (lines 303-306)

  1. Application of pyrrole copolymers, 3.2 Copolymers for bio-sensing applications: Correct “depotiion” on line 296.

Respond: It was corrected (line332)

Rewrite “…thickness of the Py and 19...” on lines 305-306.

Respond: It was corrected: “The applied potential, the composition of the co-monomer solution and the film thickness of the Py and 36 were investigated in order to obtain copolymer with specifically functionalised nanotubes with potential application in drug delivery [125].” (lines 528-539)

Are “Copolymers of pyrrole and dopamine” (line 317) really copolymers?

Respond: This sentence was changed: “In comparison with PPy-coated electrodes, polydopamine-Ppy coated electrodes exhibited both improved electrical and cell-supporting properties, the latter being evidenced by a significant improvement in the growth and differentiation of C2C12 myoblasts and PC12 neuronal cells on copolymer-coated electrodes.“ (lines 345-348)

There is no mention for Figure 6 in the text.

Respond: All figures were highlighted in the manuscript text.

Rewrite “…sensor performance fabricated own to the higher stability of obtained copolymers…” on line 333.

Respond: It was corrected: “A longer methylene moiety between osmium complex and PPy units improves the glucose sensor performance own to the higher stability of obtained copolymers.” (line 354)

  1. Application of pyrrole copolymers, 3.3 Copolymers for electrochromic applications: Rewrite “…and different in different oxidation states.”.

Respond: It was rewritten: ” EDOT significantly improves the electroactive stability of PPy derivatives and for different systems leads to better electrochromic properties in different oxidation states” (lines 470-472)

Reviewer 2 Report

This manuscript from Ledwon and coworkers aimed to summarize various polypyrrole copolymers, including their properties and applications. My major concern about current manuscript is that it now only looks like a list of "facts", which does not include critical insights on important topics, such as structure-property relationships of polypyrrole copolymers. Here are some comments and suggestions that the authors should response in the revised version:

1) In the introduction section, the mechanisms of both electro- and chemo-polymerization of polypyrrole should be discussed. A concise scheme is preferred.

2) At line 21, the authors mentioned "PPy has attracted such interest due to its excellent stability in air, biocompatibility, high conductivity and the possibility of producing it via electrochemical polymerisation in both water and organic solvents". Please provide references for supporting this statement.

3) At line 29, the authors mentioned "The first works on this class of materials have been published in the 1980s". Please provide references for supporting this statement.

4) At line 39, the authors mentioned "The conductivity of PPy is in the range 40-100 S∙cm-1, while the conductivity of poly(1) is considerably lower and approx. 10-3 S∙cm-1". Please provide references for supporting this statement.

5) At line 43, the authors mentioned "The scanning electron microscopy (SEM) micrographs reveal fundamental differences between the morphology of the homo- and copolymers." What are the differences? How are those correlated with materials' properties?

6) At line 53, the authors mentioned "The conductivity of the copolymer films were negatively correlated with both the alkyl chain length and ratio of N-alkylpyrrole units to pyrrole units in the resultant copolymers, with the highest conductivity being recorded for PPy." The authors should also include the reasoning behind these results.

7) At line 56, the authors mentioned "Homopolymers and copolymers pyrrole and 1...". This sentence should be corrected to "Homopolymers and copolymers of pyrrole and 1..."

8) At line 66, the authors mentioned "Those systems have an advantage of improved conductivity." Please include explanation if it was addressed in the original paper (Ref. 12).

9) At line 95, the authors mentioned "...and showed both higher conductivity and better thermal stability than PPy." Please also include explanation.

10) For subsection 2.2, the chemical structures of each comonomers (furan, thiophene, aniline, and carbazole) should be illustrated.

11) The authors should highlight Figure 4 in the main text.

12) The whole subsection 2.3 is a little difficult to follow. Maybe the authors could include chemical structures of those "non-conjugated" species. For example, at line 215, the authors mentioned "Electrochemically generated copolymers of Py with oligo(ethyleneoxy) chains, such as 3,6-dioxaheptyl and 3,6,9-trioxadecanyl at the 3-position...". A reaction scheme on this polymerization will definitely help readers to better understand the process.

13) The authors should highlight Figure 5 in the main text.

14) At line 269, it will be better to correct "Maldi-TOf" to "MALDI-TOF".

15) At line 276, the authors mentioned "N-alkylmethine pyrrole does not form electrochemicaly stable homopolymers, but can produce copolymers with PPy." Please also include the explanation for this behavior.

16) For subsection 3.2, since the authors claim biosensors are one of the most studied applications of polypyrrole, more figures and schemes should be included.

17) The authors should highlight Figure 5 in the main text.

18) For subsection 3.3, please briefly introduce what is electrochromic. The authors could also include one or two pictures demonstrating electrochromism of polypyrrole copolymers.

19) It is important to unify the label used in Figure 4, 5 and 6. Currently, the font size of labels (a, b, c...) are not the same.

Author Response

Thank you for your valuable comments. Please see a point-by-point response:

1) In the introduction section, the mechanisms of both electro- and chemo-polymerization of polypyrrole should be discussed. A concise scheme is preferred.

Respond: The Introduction section has been improved. We added the mechanism of electropolymerization. We added discussion. Please see a modified Introduction section.

2) At line 21, the authors mentioned "PPy has attracted such interest due to its excellent stability in air, biocompatibility, high conductivity and the possibility of producing it via electrochemical polymerisation in both water and organic solvents". Please provide references for supporting this statement.

Respond: References were added.

3) At line 29, the authors mentioned "The first works on this class of materials have been published in the 1980s". Please provide references for supporting this statement.

Respond: Reference was added.

4) At line 39, the authors mentioned "The conductivity of PPy is in the range 40-100 S∙cm-1, while the conductivity of poly(1) is considerably lower and approx. 10-3 S∙cm-1". Please provide references for supporting this statement.

Respond: Reference was added.

5) At line 43, the authors mentioned "The scanning electron microscopy (SEM) micrographs reveal fundamental differences between the morphology of the homo- and copolymers." What are the differences? How are those correlated with materials' properties?

Respond: This sentence was deleted.

6) At line 53, the authors mentioned "The conductivity of the copolymer films were negatively correlated with both the alkyl chain length and ratio of N-alkylpyrrole units to pyrrole units in the resultant copolymers, with the highest conductivity being recorded for PPy." The authors should also include the reasoning behind these results.

Respond: This part was improved: ”Copolymers of different N-alkylpyrroles (3-5) and Py are another example of new electrochemically produced materials which slight change in the monomer structure leads to significant changes in properties [25]. Copolymers and homopolymers were synthetized via both electrochemical and chemical polymerisation protocols.  Their electrical conductivity was studied in detail. The highest conductivity 0.54 S/cm was recorded for PPy. The increase in the alkyl chain length of the utilised N-alkylpyrrole co-monomer results in decreasing conductivity of the copolymer. A similar correlation was recorded for copolymers, with increasing the ratio of N-alkylpyrrole units to pyrrole units resulting in decreasing conductivity.”

7) At line 56, the authors mentioned "Homopolymers and copolymers pyrrole and 1...". This sentence should be corrected to "Homopolymers and copolymers of pyrrole and 1..."

Respond: It was corrected.

8) At line 66, the authors mentioned "Those systems have an advantage of improved conductivity." Please include explanation if it was addressed in the original paper (Ref. 12).

Respond: It can be related to the cross-linked structure.

9) At line 95, the authors mentioned "...and showed both higher conductivity and better thermal stability than PPy." Please also include explanation.

Respond: We added:” These improved properties results from the enhanced conjugation in the copolymer main chain.”

10) For subsection 2.2, the chemical structures of each comonomers (furan, thiophene, aniline, and carbazole) should be illustrated.

Respond: The structures of comonomers were added. Please see Figure 4.

11) The authors should highlight Figure 4 in the main text.

Respond: All figures were highlighted in the manuscript text.

12) The whole subsection 2.3 is a little difficult to follow. Maybe the authors could include chemical structures of those "non-conjugated" species. For example, at line 215, the authors mentioned "Electrochemically generated copolymers of Py with oligo(ethyleneoxy) chains, such as 3,6-dioxaheptyl and 3,6,9-trioxadecanyl at the 3-position...". A reaction scheme on this polymerization will definitely help readers to better understand the process.

Respond: We modified this part. Part of text was moved to subsections 2.1 and new subsections 2.4

13) The authors should highlight Figure 5 in the main text.

Respond: All figures were highlighted in the manuscript text.

14) At line 269, it will be better to correct "Maldi-TOf" to "MALDI-TOF".

Respond: It was corrected

15) At line 276, the authors mentioned "N-alkylmethine pyrrole does not form electrochemicaly stable homopolymers, but can produce copolymers with PPy." Please also include the explanation for this behavior.

Respond: We modified this part.

16) For subsection 3.2, since the authors claim biosensors are one of the most studied applications of polypyrrole, more figures and schemes should be included.

Respond: We added Figure 6.

17) The authors should highlight Figure 5 in the main text.

Respond: All figures were highlighted in the manuscript text.

18) For subsection 3.3, please briefly introduce what is electrochromic. The authors could also include one or two pictures demonstrating electrochromism of polypyrrole copolymers.

Respond: The introduction to the subsection 3.3 was added. Figure 6 was added.

19) It is important to unify the label used in Figure 4, 5 and 6. Currently, the font size of labels (a, b, c...) are not the same.

Respond: The font size off all Figures was unified.

Reviewer 3 Report

The article 'Electrochemically produced copolymers of pyrrole 2 and its derivatives: A plentitude of material 3 properties using “simple” heterocyclic co-monomers' is a review that addresses the electrochemical synthesis and applications of homo- and copolymers of pyrrole.

The subject of the review is of interest for a readership of the Materials journal. The article can be published after making some corrections to improve the quality of the manuscript.

The main comment concerns certain informality of the text.

Section 2, devoted to the synthesis of (co)polymers, does not contain the reaction schemes of polymerization, however, this section contain fragmentary discussion of the electroconductivity of the polymers obtained. I strongly recommend the addition of the reaction schemes to show the chemical structures of different (co)polymers.

Might be a good idea to divide the review by types of (co)polymers, with simultaneous discussion of (co)polymer characteristics and applications.

Minor comments:

lines 4, 28 and below – comonomers, not co-monomers

line 40 and below – please consider the difference between - (hyphen) and – (dash)

lines 45, 87, 126 – it might be better to combine Figures 1, 2 and 3 with the same figure captions. Alternatively, if it was decided to reorganize the review by the types of (co)polymers, corresponding Figure should be added at the beginning of each Section.

lines 141–145 – the caption of the Figure 4 should be changed by the addition of the naming of copolymer at the beginning of the caption.

Author Response

The main comment concerns certain informality of the text.

Respond: We improved the manuscript text in many places.

Section 2, devoted to the synthesis of (co)polymers, does not contain the reaction schemes of polymerization, however, this section contain fragmentary discussion of the electroconductivity of the polymers obtained. I strongly recommend the addition of the reaction schemes to show the chemical structures of different (co)polymers.

Respond: We added Schemes. Scheme 1 show the pyrrole polymerisation mechanism. Scheme 2 show possible copolymer structures. We also added discussion to the Introduction Section. Please see a modified Introduction section.

Might be a good idea to divide the review by types of (co)polymers, with simultaneous discussion of (co)polymer characteristics and applications.

Respond: We modified Sections 2 and 3. Additional subsections were added to both Sections in order to increase the readability of the manuscript.

 Minor comments:

 lines 4, 28 and below – comonomers, not co-monomers

Respond: We use British English in which co-monomers is the preferred form.

line 40 and below – please consider the difference between - (hyphen) and – (dash)

Respond: We corrected it.

lines 45, 87, 126 – it might be better to combine Figures 1, 2 and 3 with the same figure captions. Alternatively, if it was decided to reorganize the review by the types of (co)polymers, corresponding Figure should be added at the beginning of each Section.

Respond: We reorganize these Figures by the types of monomers. More monomer structures are shown in the revised manuscript.

lines 141–145 – the caption of the Figure 4 should be changed by the addition of the naming of copolymer at the beginning of the caption.

Respond: Figure 4 has been replaced by other figures.

Round 2

Reviewer 2 Report

Thank you for addressing my previous comments and questions. Here is just one minor suggestion:

1) In Scheme 2, the structure of the random copolymers should be redrawn. Current structure only have one alkyl pyrrole unit in the middle of the chain.

Author Response

1) In Scheme 2, the structure of the random copolymers should be redrawn. Current structure only have one alkyl pyrrole unit in the middle of the chain.

Respond: We corrected Scheme 2. Please see corrected Scheme in the manuscript.